# Are There Any Red Flag Injuries in Severely Injured Patients in Older Age?

**DOI:** 10.3390/jcm10020185

**Published:** 2021-01-07

**Authors:** Daniel Popp, Borys Frankewycz, Siegmund Lang, Antonio Ernstberger, Volker Alt, Michael Worlicek, Maximilian Kerschbaum

**Affiliations:** 1Department of Trauma Surgery, University Medical Centre Regensburg, 93053 Regensburg, Germany; borys.frankewycz@ukr.de (B.F.); siegmund.lang@ukr.de (S.L.); volker.alt@ukr.de (V.A.); michael.worlicek@ukr.de (M.W.); maximilian.kerschbaum@ukr.de (M.K.); 2Clinic for Trauma and Orthopedic Surgery, University Medical Centre Regensburg, Franz-Josef-Strauss Allee 11, 93053 Regensburg, Germany; 3Department of Trauma and Hand Surgery, Hospital Osnabrück, 49076 Osnabrück, Germany; antonio.ernstberger@ukr.de

**Keywords:** severe injury, elderly patients, head injury, mortality

## Abstract

Introduction: Severely injured elderly patients pose a significant burden to trauma centers and, compared with younger patients, have worse prognoses and higher mortality rates after major trauma. The objective of this study was to identify the etiological mechanisms that are associated with severe trauma in elderly patients and to detect which injuries correlate with high mortality in elderly patients. Methods: Using a prospect cohort study model over an 11-year period, severely injured patients (ISS ≥ 16) were divided into two age groups (Group 1: 18–64; Group 2: 65–99 years). A comparison of the groups was conducted regarding injury frequency, trauma mechanism, distribution of affected body parts (AIS and ISS regions) and injury related mortality. Results: In total, 1008 patient were included (Group 1: *n* = 771; Group 2: *n* = 237). The most relevant injury in elderly patients was falling from low heights (<3 m) in contrast to traffic accident in young patients. Severely injured patients in the older age group showed a significantly higher overall mortality rate compared to the younger group (37.6% vs. 11.7%; *p* = 0.000). In both groups, the 30-day survival for patients without head injuries was significantly better compared to patients with head injuries (92.7% vs. 85.3%; *p* = 0.017), especially analyzing elderly patients (86.6% vs. 58.6%; *p* = 0.003). The relative risk of 30-day mortality in patients who suffered a head injury was also higher in the elderly group (OR: Group 1: 4.905; Group 2: 7.132). Conclusion: In contrast to younger patients, falls from low heights (<3 m) are significant risk factors for severe injuries in the geriatric collective. Additionally, elderly patients with an ISS ≥ 16 had a significantly higher mortality rate compared to severe injured younger patients. Head injuries, even minor head traumata, are associated with a significant increase in mortality. These findings will contribute to the development of more age-related therapy strategies in severely injured patients.

## 1. Introduction

As the proportion of people 65 years and older is rising and is expected to grow in Europe to at least 30% by 2050, a concomitant increase in polytrauma in the geriatric sector is predicted [1]. Accompanied by a higher level of activity, the number of older patients with severe injuries is increasing [2,3], therefore trauma surgery facilities have to integrate strategies that address these demographic changes.

In contrast to the younger population, elderly patients experience significantly higher mortality rates, complications and worse prognoses after major trauma [4,5,6,7]. Additionally, elderly patients are more likely to suffer from low-energy trauma and therefore their injuries may be underestimated [8]. As a consequence, elderly patients are more likely to be treated with a “wait and see” attitude [9].

Since the widely accepted standardized trauma scores do not unequivocally apply to elderly patients, injury mechanisms and outcome parameters need to be re-evaluated regarding their relevance in the elderly population [10].

In this study, two age-related questions were addressed by prospectively investigating severely injured patients:What are the etiological mechanisms that are associated with severe trauma in elderly patients?Which affected body parts are associated with high mortality in elderly patients?

## 2. Methods

A prospective cohort study was chosen to investigate relevant differences between two age groups regarding trauma mechanisms and factors correlated with mortality of severely injured patients. Patients who were admitted to our emergency department (Level 1 trauma) between 2006 and 2017 were recruited for the study. Inclusion criterion was an ISS (Injury Severity Score) ≥ 16 [11]. Exclusion criteria were an ISS < 16 and age < 18 years. Data collection was managed by study assistants (24 h/7 days) who were included in the treatment decision algorithm. The study was approved by the institutional ethical review board (14-101-0004).

The included patients were divided into two groups and compared to each other (Group 1: 18–64 years; Group 2: 65–99 years). The demographic data are shown in Table 1. Age, gender, ISS, length of hospital stay, injury pattern and trauma mechanisms were recorded. Injury severity was assessed with the AIS (Abbreviated Injury Score) [12]. Groups were compared regarding ISS body regions: head, face, chest, abdominal/pelvis, extremities and external/other. All injuries with an AIS ≥ 1 were included in the evaluation.

In the next step, the 30-day mortality rate in both groups was analyzed and a special focus was set on the relevance of head injuries.

Finally, injury entities were identified regarding the highest mortality in both groups.

### Statistical Analysis

Kaplan–Meier curves were used to detect differences in survival rates of both groups. A univariate data analysis was performed to compare the two age groups. The Chi-Square-Test (x^2^-Test) was used to analyze binary or nominal target variables. Logistic regression analyses with the target variable “30-day-mortality” followed. The influence of severe injured body parts on lethality was analyzed. All influence factors were included in a logistic regression analysis (backward elimination). *p*-values and odds ratios (OR) for each factor were calculated as well as the corresponding 95% confidence intervals (CI). The statistical analysis (level of significance, *p* < 0.05) was carried out using SPSS software (SPSS Inc., Chicago, IL, USA).

## 3. Results

### 3.1. Demographic Data

In total, 1008 patients met the inclusion criteria (Group 1: 771; Group 2: 237). In both groups, the majority of patients were male. The average age in Group 1 was 38.5 ± 14.2 years, and 76.5 ± 7.0 years in Group 2. The average ISS was almost identical in both groups with 31.6 ± 14.3 and 32.5 ± 17.1 (*p* = 0.470), respectively.

Patients in Group 1 had a longer hospital stay of 20.3 ± 16.5 days compared to patients in Group 2 with 16.6 ± 16.3 days (*p* = 0.002).

### 3.2. Mechanism

The main causes of severe trauma in Group 1 were car accidents, followed by motorcycle accidents and in decreasing order: falls from great heights (≥3 m), others, falls from moderate heights (<3 m), bicycle falls and accidents as pedestrians (Figure 1).

In contrast, the most frequent trauma mechanism in Group 2 were (in decreasing order): fall from <3 m, car accidents, bicycles accidents, falls from ≥3 m, pedestrian accidents, others and motorbike accidents (Figure 1).

### 3.3. ISS Body Regions

In general, the most common AIS ≥ 1 injuries in both groups were found in the ISS regions head, chest and extremities. The head was the only region that had a higher prevalence in Group 2 than in Group 1 of the severely injured patients (Figure 2). Overall, 85.7% of the elderly patients in Group 2 suffered from head injuries compared to 71.6% of the younger patients in Group 1 (Figure 2a). The face was affected in 31.5% of Group 1 and in 21.1% of Group 2 patients (Figure 2b). Injuries in the chest area were found in 74.2% of Group 1 and in 58.2% of Group 2 patients (Figure 2c). Abdominal/pelvic injuries appeared in 46.8% in Group 1 and in 29.5% in Group 2 patients (Figure 2d). Extremity injuries were prevalent in 69.3% of Group 1 and in 54.0% of Group 2 patients (Figure 2e). Finally, external/other injuries were found in 55.5% of Group 1 and in 50.6% of Group 2 patients (Figure 2f).

### 3.4. Thirty Day Survival/Mortality Rate

Overall, Group 1 had a 30-day mortality rate of 11.7%, whereas Group 2 had a significantly higher 30-day mortality rate of 37.6% (*p* = 0.003) (Figure 3). The 30-day survival for patients in both groups without head injuries was significantly better (92.7% vs. 85.3%) (*p* = 0.017) (Figure 4A) compared to patients with head injuries. In particular, elderly patients (Group 2) with head injuries in particular showed a relevant decrease of the 30-day survival (86.6% vs. 58.6%) (*p* = 0.003) (Figure 4B).

### 3.5. Influence of Head Injuries on the Mortality Rate

When analyzing the influence of the severity of head injuries (AIS 0 to AIS 6) during the 30 days of survival, the mortality rate for minor to severe head injuries varied from 0.0% to 7.3% in Group 1 (AIS 0: 7.3%; AIS 1: 0.0%; AIS 2: 2.2%;, AIS 3: 1.5%; AIS 4: 5.7%) and from 9.1% and 26.7% in Group 2 (AIS 0: 14.7%; AIS 1: 9.1%; AIS 2: 26.7%; AIS 3: 26.1%; AIS 4: 19.6%). Critical head injuries (AIS 5) led to an increased mortality rate of 28.7% in Group 1 and 61.4% in Group 2 (Figure 5).

The relative risk (odds ratio) to die of head injuries was 45% higher in the older population (7.132 vs. 4.905, Table 2) compared to the younger patients.

The odds ratio for abdominal injuries shows a similar distribution (2.939 vs. 1.851, Table 2), but it is overall noticeably lower than that for head injuries. In all other body regions, no significant differences can be shown.

## 4. Discussion

In this prospective cohort study, the mechanisms and injury patterns of severe trauma in elderly people were investigated. The key findings of this study were that primarily falls from low heights (<3 m) lead to relevant severe trauma and that head injuries have the biggest impact on the mortality rate in elderly patients.

Treatment of severely injured patients in trauma surgery is an increasing challenge in everyday life. In recent years, development of algorithms has noticeably improved the outcome [13,14,15]. However, the change in demographics creates new challenges for the clinical routine. The number of older patients who are admitted to emergency rooms with severe injuries is continuously rising [16,17,18]. To initiate diagnostics and therapy strategies as efficiently and purposefully as possible, it is necessary to identify specific injury patterns and associated mortality rates to improve the outcome of this highly vulnerable group of elderly patients [19].

Interestingly, in contrast to younger patients, where road traffic accidents are the most common reason for severe trauma, the main trauma mechanism in elderly patients is falling from low heights < 3 m. In contrast to the younger population, road traffic accidents played a subordinate role in the elderly patient group. Comparable data were found by Lowe et al., who reported a large increase of falling accidents as a main reason for severe trauma in elderly patients [20].

Data from this study confirm the results of previous studies, which showed that severely injured elderly patients have a significant higher mortality rate than younger patients [5,18,21].

In this study, the total mortality rate of the elderly patients was three times higher than in the younger group. In particular, it was shown, that head injuries had the biggest impact on the mortality rate in elderly patients. This effect occurred even with minor head injuries, which is in contrast to the younger patients, who only had a significant higher mortality rate beginning with a head injury (AIS 5). The combination of increased frequency and mortality underlines the vulnerability of this specific patient group.

In a recent study, Beedham et al. described that frail, older patients often sustain head injury when they fall and are predisposed to hemorrhagic complications because of anticoagulant use and the effects of aging [22]. This is in accordance with the findings of Karibe et al., who found falling to be one of the most common reasons for traumatic brain injury, caused by degraded motoric and physiological functions [23].

There are several reasons older people are at risk of suffering intracranial bleeding following a fall. Population level data suggest that prescription of anticoagulation increases the risk of intracranial bleeding [24] and antithrombotic therapy is becoming common among seniors [25,26]. Additionally, falling in itself is associated with frailty [27] and especially seniors in a poorer health state are most likely to fall and, therefore, may have a higher risk of intracranial bleedings [28].

The different effect of head injuries in younger and older patients implies a different significance of primary and secondary injuries in these groups. It also indicates that aging, comorbidities, medications and deficiency of rehabilitation potential substantially contribute to poor prognosis [29].

Another aspect of the increased mortality of elderly patients are therapy limitations and the fact that the inert will to live can be diminished. This is accompanied by corresponding restrictions, e.g., the refusal of intensive care measures such as parenteral nutrition or antibiotic treatment [30,31]. Such restrictions are rarely found in the group of young patients.

A study of more than 22,500 trauma patients (including more than 7100 geriatric trauma patients) revealed that geriatric patients had significantly lower intensive care unit (ICU) admission rates compared with younger patients with similar injury severity [32]. Improving triage by placing appropriate geriatric patients in the ICU may be the first step in morbidity and mortality improvement.

A team approach with interdisciplinary care involving geriatricians, social workers and pharmacists supervised by surgeons could improve the quality of trauma care to address comorbidities, geriatric syndromes, care planning and rehabilitation. A targeted patient management including the optimal medication and pain management right from the beginning of hospitalization may reduce mortality and improve functional outcome [33,34,35].

Certainly, the study has several limitations. First, data that differentiate between the comorbidities of patients were not taken into account and could give a more detailed insight whether there is a triggering pre-existing condition as well. Secondly, the medication in the patients’ history was not evaluated. However, this study gives significant insights into key etiological and outcome aspects that need to be analyzed in following studies.

## 5. Conclusions

Elderly patients with an ISS ≥ 16 and particularly those with head injuries have a significant higher mortality rate. Head injuries should be regarded as red flags when treating severe injured elderly patients, regardless of their severity.

## Figures and Tables

**Figure 1 jcm-10-00185-f001:**
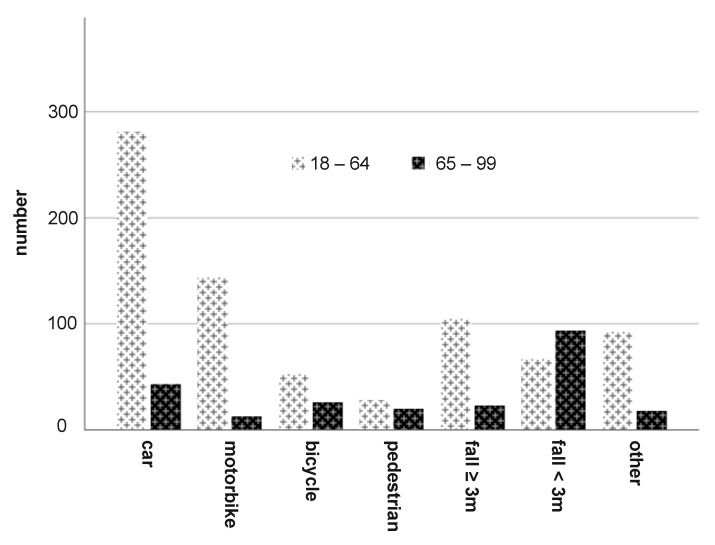
Trauma mechanism in both age groups.

**Figure 2 jcm-10-00185-f002:**
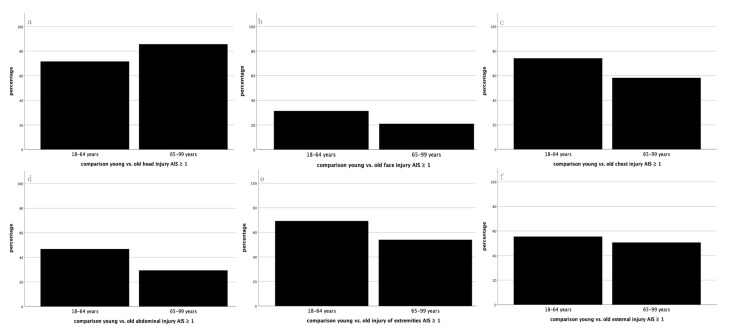
Percentage distribution of prevalent injuries by ISS body region: (**a**) head; (**b**) face; (**c**) chest; (**d**) abdomen/pelvis; (**e**) extremities; and (**f**) external/other. The black bar represents AIS ≥ 1 injuries.

**Figure 3 jcm-10-00185-f003:**
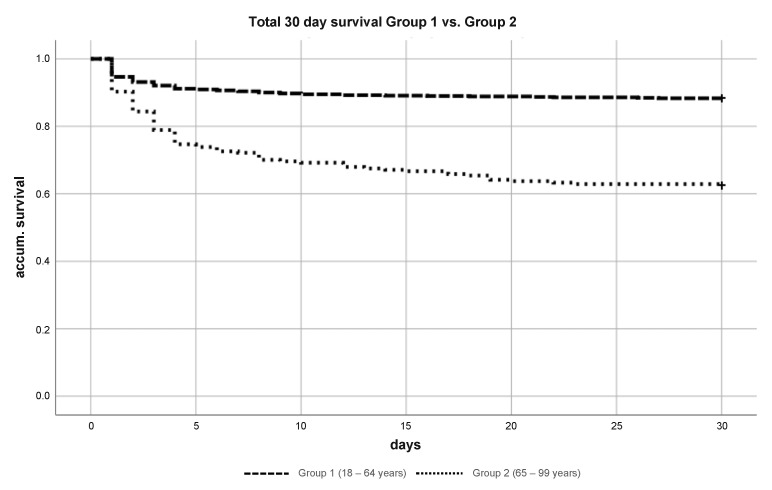
Kaplan–Meier curve of 30-day survival severely injured patient (ISS ≥ 16) Group 1 vs. Group 2.

**Figure 4 jcm-10-00185-f004:**
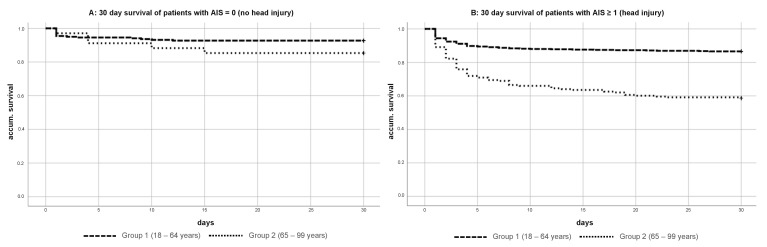
(**A**,**B**) Kaplan–Meier curve of 30-day survival severely injured patient (ISS ≥ 16) Group 1 vs. Group 2 with and without head injury AIS ≥ 1.

**Figure 5 jcm-10-00185-f005:**
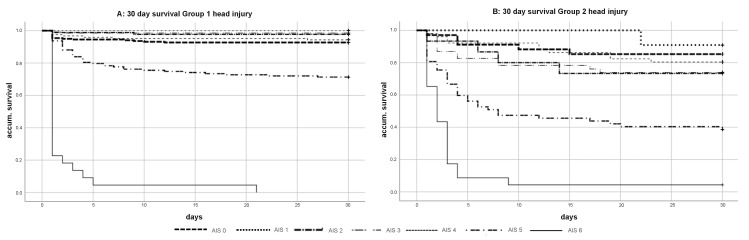
(**A**,**B**) Kaplan–Meier curves of 30-day survival in dependence of the head severity (with AIS 0 = no head injury and AIS 6 = maximum head injury).

**Table 1 jcm-10-00185-t001:** Demographic data of analyzed patients.

	Age Group 18–64 Years	Age Group 65–99 Years	*p*-Values
Number (n)	771	237	
Male (n/%)	599/77.7	147/62	0.000 *
Age (years ± SD)	38.5 ± 14.2	76.5 ± 7.0	
ISS (Ø ± SD)	31.6 ± 14.3	32.5 ± 17.1	0.470
Length of hospital stay (days ± SD)	20.3 ± 16.5	16.6 ± 16.3	0.002 *

* significant.

**Table 2 jcm-10-00185-t002:** Odds ratios of ISS injury regions in both age groups.

	SD	Sig.	Odds Ratio		SD	Sig.	Odds Ratio
Head trauma AIS ≥ 1	0.295	0.000	**4.905 ***	Head trauma AIS ≥ 1	0.479	0.000	**7.132 ***
Face trauma AIS ≥ 1	0.320	0.063	**1.813**	Face trauma AIS ≥ 1	0.604	0.545	**1.442**
Chest trauma AIS ≥ 1	0.260	0.001	**2.311 ***	Chest trauma AIS ≥ 1	0.294	0.668	**1.134**
Abdominal trauma AIS ≥ 1	0.286	0.032	**1.851 ***	Abdominal trauma AIS ≥ 1	0.542	0.047	**2.939 ***
Extremety trauma AIS ≥ 1	0.247	0.879	**1.038**	Extremety trauma AIS ≥ 1	0.361	0.266	**1.494**
External injury AIS ≥ 1	0.622	0.046	**3.466 ***	External injury AIS ≥ 1	0.883	0.271	**2.644**
Odds ratio age group 18–64				Odds ratio age group 65–99			

* significant.

## Data Availability

Data sharing not applicable. Access to the source dataset is only permitted to employees of Department of Trauma Surgery, University Medical Centre Regensburg.

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
