# Peer review of "Are There Any Red Flag Injuries in Severely Injured Patients in Older Age?"

_jcm, 2021, doi:10.3390/jcm10020185_

Round 1

Reviewer 1 Report

Overall a beautiful article very useful for the decision making of any neurosurgeon.

Few details to review:

-The type of head injury should be better specified. In elderly patients it changes a lot if the trauma has caused extracerebral or intracerebral haematomas. In an elderly patient with comorbidities it is worthwhile to operate if there is a extracerebral haematoma rather than operating for intraparenchymal which, as can be seen from the study in question, would not obtain advantages from the surgery.

A recent study published in the new England Journal of Medicine show that eldery patients with cronic subdural haematoma should be have an operation to drain the haematoma and the conservative treatment with dexamethasone shoul be avoided.

-Furthermore, I would better specify the conclusions ,specifying the red flags based on the collected data.

Author Response

Dear reviewer,

We very much appreciate the thoughtful and insightful comments you made and questions put forth by each of the reviewers, and have provided a point-by-point response to these below and have revised the submission to include these excellent points where appropriate.

We were delighted to hear that, pending appropriate revision, the above noted manuscript might be suitable for publication in JCM as a Research Article.

Thanks very much for your input and efforts towards improving our work. 

Best regards,

The authors

Reviewer 1

Overall a beautiful article very useful for the decision making of any neurosurgeon.

Few details to review:

-The type of head injury should be better specified. In elderly patients it changes a lot if the trauma has caused extracerebral or intracerebral haematomas. In an elderly patient with comorbidities it is worthwhile to operate if there is a extracerebral haematoma rather than operating for intraparenchymal which, as can be seen from the study in question, would not obtain advantages from the surgery.

A recent study published in the new England Journal of Medicine show that elderly patients with cronic subdural haematoma should be have an operation to drain the haematoma and the conservative treatment with dexamethasone should be avoided.

-Furthermore, I would better specify the conclusions, specifying the red flags based on the collected data.

Response:

Thank you very much for your substantial input. Unfortunately this further specification of head traumata was not part of our study. However, we agree with you that further investigations will yield additional clinically important findings. For this reason, we are currently working on a pursuing  investigation, but the results are not yet available.

Reviewer 2 Report

Manuscript ID: jcm-1063322 Are there any red flag injuries in severely injured patients in older age?

Severely injured elderly patients pose a significant burden to trauma centres, and compared with younger patients, have worse prognoses and higher mortality rates after major trauma.

The authors aimed to identify the etiological mechanisms associated with severe trauma in elderly patients and to detect which injuries were correlated with a higher mortality in elderly patients.

This was a prospect cohort study model of 1008 severely injured patients (ISS ≥ 16) patients over an 11-year period. The patients were divided into two age groups (group 1: 18-64 771 patients; group 2: 65-99 years 237 patients), and were compared for injury frequency, trauma mechanism, distribution of affected body parts (AIS and ISS regions) and injury related mortality.

In general, elderly patients fell from low heights (< 3 meters); younger patients were involved with traffic accidents. Severely injured patients(ISS ≥ 16) in the older age group showed a significant higher overall mortality rate compared to the younger group (37.6% vs. 11.7%;  p=0.000). The 30-day survival for patients without head injuries was significant better compared to patients with head injuries (92.7% vs 85.3%; p=0.017), particularly in the elderly (86.6% vs. 58.6%; p=0.003). The relative risk of 30-day mortality in patients who suffered a head injury was also higher in the elderly group (OR: group 1: 4.905; group 2: 7.132).

The authors conclude that falls from low heights (< 3 meters) were a significant risk factors for severe injuries in elderly patients, and that those who suffered more severe injuries and head injuries (even minor head injuries) had a significantly higher mortality rate.

The authors went on the suggest that their findings would contribute to the development of more age-related therapy strategies in severely injured patients.

This paper has not really added much to our body of data in the literature. It is well known that the elderly suffer frequent low energy injuries, usually from standing height, and that their outcomes are generally worse than younger patients following any level of trauma. I think that the observation that head injuries pose a particularly increased risk of mortality is useful.

Generally speaking, I do like papers with data, and all data is helpful in building a fuller picture.

I have some comments

1)

The authors have analysed very broad age groups, and with this number of patients it would have been more helpful to looks at other parameters such as patient co-morbidities, BMI, ASA, use of anticoagulants, stratification by tighter age ranges.

Would they care to comment on why this was not done, and maybe highlight these deficiencies in their analyses?

2)

The authors concluding statement in their abstract is rather nonsensical.

“Falls from low heights (< 3 meters) are significant risk factors for severe injuries in the geriatric collective”.

This actually includes every single fall. In other words in order to prevent the elderly from suffering severe injuries, one would really have to prevent ever single fall!

3)

In the abstract the other concluding remark is not altogether helpful.

They suggested that their findings “would contribute to the development of more age-related therapy strategies in severely injured patients”.

This is an epidemiological paper examining the mortality following trauma, rather than a clinical paper examining the impacts of a treatment/intervention. Ie: this is an observation of patients after these event. Thus therapy strategies would not have any influence on factors such as the height from which the patient fell, nor other elements of their mechanism of injury.

4)

I think that the conclusion in the Abstract should match what the authors have out in their main manuscript.

Something like this.

“Elderly patients with an ISS ≥ 16 and particularly those with head injuries have a significant higher mortality rate. Head injuries should be regarded as red flags when treating severe injured elderly patients, regardless of their severity.”

5)

In the abstract introduction, please remove “complications rates”. As this paper has not looked at complications per se.

Maybe something like,

“Severely injured elderly patients pose a significant burden to trauma centres, and compared with younger patients, have worse prognoses and higher mortality rates after major trauma.”

6)

The written English could be improved throughout the manuscript.

Author Response

Dear reviewer,

We very much appreciate the thoughtful and insightful comments you made and questions put forth by each of the reviewers, and have provided a point-by-point response to these below and have revised the submission to include these excellent points where appropriate.

We were delighted to hear that, pending appropriate revision, the above noted manuscript might be suitable for publication in JCM as a Research Article.

Thanks very much for your input and efforts towards improving our work. 

Best regards,

The authors

Reviewer 2

Severely injured elderly patients pose a significant burden to trauma centres, and compared with younger patients, have worse prognoses and higher mortality rates after major trauma.

The authors aimed to identify the etiological mechanisms associated with severe trauma in elderly patients and to detect which injuries were correlated with a higher mortality in elderly patients.

This was a prospect cohort study model of 1008 severely injured patients (ISS ≥ 16) patients over an 11-year period. The patients were divided into two age groups (group 1: 18-64 771 patients; group 2: 65-99 years 237 patients), and were compared for injury frequency, trauma mechanism, distribution of affected body parts (AIS and ISS regions) and injury related mortality.

In general, elderly patients fell from low heights (< 3 meters); younger patients were involved with traffic accidents. Severely injured patients(ISS ≥ 16) in the older age group showed a significant higher overall mortality rate compared to the younger group (37.6% vs. 11.7%;  p=0.000). The 30-day survival for patients without head injuries was significant better compared to patients with head injuries (92.7% vs 85.3%; p=0.017), particularly in the elderly (86.6% vs. 58.6%; p=0.003). The relative risk of 30-day mortality in patients who suffered a head injury was also higher in the elderly group (OR: group 1: 4.905; group 2: 7.132).

The authors conclude that falls from low heights (< 3 meters) were a significant risk factors for severe injuries in elderly patients, and that those who suffered more severe injuries and head injuries (even minor head injuries) had a significantly higher mortality rate.

The authors went on the suggest that their findings would contribute to the development of more age-related therapy strategies in severely injured patients.

This paper has not really added much to our body of data in the literature. It is well known that the elderly suffer frequent low energy injuries, usually from standing height, and that their outcomes are generally worse than younger patients following any level of trauma. I think that the observation that head injuries pose a particularly increased risk of mortality is useful.

Generally speaking, I do like papers with data, and all data is helpful in building a fuller picture.

I have some comments

1)

The authors have analysed very broad age groups, and with this number of patients it would have been more helpful to looks at other parameters such as patient co-morbidities, BMI, ASA, use of anticoagulants, stratification by tighter age ranges.

Would they care to comment on why this was not done, and maybe highlight these deficiencies in their analyses?

Response:

Thank you very much for this important point. With regard to the categorisation and the formulation of the special issue "elderly patients", we have followed the categorisation frequently used in the literature. Here, elderly patients are defined as being 65 years and older. The breakdown of co-morbidities was not part of our evaluation and was not taken into account in the initial data collection. This is mentioned in the limitations (line 191-195). Further studies are planned to address this issue.

2)

The authors concluding statement in their abstract is rather nonsensical.

“Falls from low heights (< 3 meters) are significant risk factors for severe injuries in the geriatric collective”.

This actually includes every single fall. In other words in order to prevent the elderly from suffering severe injuries, one would really have to prevent ever single fall!

Response:

Thank you for this critical question. From our point of view, this is exactly what needs to be considered in clinical routine. Even supposed low-energy traumas can, as could be interpreted from our data, result in more severe injuries in contrast to young patients.

However, this does not mean that every fall of an elderly person should be treated in the same way as a serious traffic accident. Though, the treating physician should keep in mind that minor traumas can also lead to serious injuries in the elderly.

3)

In the abstract the other concluding remark is not altogether helpful.

They suggested that their findings “would contribute to the development of more age-related therapy strategies in severely injured patients”.

This is an epidemiological paper examining the mortality following trauma, rather than a clinical paper examining the impacts of a treatment/intervention. Ie: this is an observation of patients after these event. Thus therapy strategies would not have any influence on factors such as the height from which the patient fell, nor other elements of their mechanism of injury.

Response:

Thanks for this observation. In our paper, we studied severely injured patients with an ISS ≥ 16. This allows us to name different trauma mechanisms that lead to severe injuries in the respective age groups. In addition, we were able to demonstrate the links between region of injury and mortality. An important result here is, that even supposedly minor head injuries in elderly patients are associated with increased mortality in the context of a severely injured patient with multiple injuries. Based on this fact, clinical action should be sharpened and adapted in order to possibly achieve a reduction in mortality in the future. This was not investigated in our study but should serve as an outlook for future studies.

4)

I think that the conclusion in the Abstract should match what the authors have out in their main manuscript.

Something like this.

“Elderly patients with an ISS ≥ 16 and particularly those with head injuries have a significant higher mortality rate. Head injuries should be regarded as red flags when treating severe injured elderly patients, regardless of their severity.”

Response:

Thank you very much for this helpful comment. We changed the conclusion in our manuscript according to your sharpened proposal.

5)

In the abstract introduction, please remove “complications rates”. As this paper has not looked at complications per se.

Maybe something like,

“Severely injured elderly patients pose a significant burden to trauma centres, and compared with younger patients, have worse prognoses and higher mortality rates after major trauma.”

Response:

Thanks for the effort you put in this section. We adapted the introduction corresponding to your suggestion.

6)

The written English could be improved throughout the manuscript.

Response:

Thank you for this note. We had the text checked and corrected by a native speaker.